# Pap Smear Cancer Coverage in Brazilian Capitals including the Pandemic Period Caused by the SARS-CoV-2 Virus: Ecological Study

**DOI:** 10.3390/ijerph21030303

**Published:** 2024-03-05

**Authors:** Annielson de Souza Costa, Edmund Chada Baracat, José Maria Soares Júnior, Isabel Cristina Esposito Sorpreso

**Affiliations:** Laboratorio de Ginecologia Estrutural e Molecular (LIM-58), Disciplina de Ginecologia, Departamento de Obstetricia e Ginecologia, Hospital das Clinicas HC-FMUSP, Faculdade de Medicina, Universidade de Sao Paulo, Sao Paulo 05403-010, Brazil; ecbaracat@usp.br (E.C.B.); jose.msjunior@fm.usp.br (J.M.S.J.); icesorpreso@usp.br (I.C.E.S.)

**Keywords:** cervical cancer, health information systems, health indicators

## Abstract

Introduction: Cervical cancer develops slowly and may not manifest signs and symptoms at an early stage. It is worth mentioning the factors that can influence the onset of cervical cancer: smoking, early sexual life, multiple sexual partners, use of oral contraceptives, multiparity, low socioeconomic status, among others. An important risk factor for the onset of this disease is HPV infection, a virus associated with most cases of precursor lesions of this type of cancer. It is essential to understand the comprehensiveness of the scope and adherence to the recommended guidelines throughout the national territory. Therefore, health indicators are important management tools that make it possible to evaluate the services offered, measuring the reach of the target population, the supply and access to preventive exams. Objective: To analyze the trend of progress in cervical cancer coverage actions in Brazilian capitals from 2016 to 2021. Method: This is an ecological study with temporal analysis that used secondary data referring to process indicators for cervical cancer control actions in women aged between 25 and 64 years living in Brazilian capitals between 2016 and 2021. Aspects related to the quality of care in the cervical cancer prevention program were evaluated using databases of the Cancer Information System (SISCAN), available in DATASUS. The indicators used to monitor and evaluate cervical cancer control actions were (i) cervical coverage, (ii) reason for cervical surgery, and (iii) proportion of cervical cancer every 3 years. Results: In 2016, 410,000 tests were performed and notified in the SISCAN system in all Brazilian capitals, with emphasis on Curitiba, with 65,715 tests performed, and Porto Velho, with 174. In 2020, there was a reduction in exams compared to the previous year in all capitals, with the exception of Palmas, which went from 7655 exams to 9604. It was observed that all the capitals studied showed an increase in the annual percentage variation of Pap smear coverage, with the exception of Brasília, Manaus, Porto Alegre and Porto Velho, which did not show a statistically significant increase (APC = 3.01, 2.746, 3.987, 3.69, respectively). When analyzing the performance of oncotic cytology exams in the capitals according to the years 2019 and 2020, it was observed that only Manaus registered an increase in the number of procedures performed, reaching a difference of 56.5% from one year to the next. Conclusion: The ecological analysis revealed a worrying drop in the number of tests performed in 2020, reflecting a sharp drop in coverage actions in Brazilian capitals during the pandemic caused by the SARS-CoV-2 virus. The pandemic has exacerbated existing inequalities and highlighted the need for adaptive strategies to maintain essential screening services in times of crisis.

## 1. Introduction

Cervical cancer is a condition that develops slowly and may not have signs and symptoms at an early stage. It can be caused by the human papillomavirus (HPV), which is transmitted primarily through sexual intercourse. Persistent infection by oncogenic types of the human papillomavirus (HPV) is a determining factor in the appearance of this cancer, with types 16 and 18 accounting for about 70% of cases [1,2].

The natural history of the disease has a slow development, screening is an essential tool for early detection, as it allows the recognition of precursor lesions that can be treated and cured before they can progress to cancer [2,3]. The Pap smear is the most widely used screening method in Brazil. It should be performed on women aged 25 to 64 years who have already started sexual life for 2 years in a row, if both results are negative; it is now held every 3 years [4].

Among the factors that can influence the onset of cervical cancer are smoking, the onset of sexual life, the multiplicity of sexual partners, the use of oral contraceptives, multiparity, and low socioeconomic status. An important risk factor for the onset of this disease is HPV infection, a microorganism associated with most cases of precursor lesions of this type of cancer. Pap smears can detect early lesions of this type [5].

These factors, when well managed, can trigger an early diagnosis of the disease. When this diagnosis does not occur, and cervical cancer is discovered late, mortality rates in the Brazilian population become high. Diagnosis should occur promptly for appropriate treatment, thus increasing the possibility of cure [6].

An important prevention device against HPV is immunization through the vaccine, which was added to the national vaccination schedule of the SUS in March 2014. Currently, the HPV vaccine is available for girls aged 9 to 14 years and boys aged 11 to 14 years, people aged 9 to 45 years with HIV/AIDS who have undergone organ or bone marrow transplantation, and cancer patients who have weakened immune systems [7]. The vaccination schedule consists of two doses, administered 6 months apart [6].

It is recommended that the interval between doses should not exceed 12 to 15 months to ensure high antibody production and vaccination efficacy. However, if adolescents or young people are late with vaccination doses, even if the recommended interval of 12 to 15 months has passed, they should continue the vaccination schedule when they go to the vaccination rooms, without the need to restart [7].

The Pap smear is considered effective. Women who undergo the test have a greater possibility of early diagnosis, reducing the number of affected individuals in advanced stages of the disease and increasing the chances of treatment. Coverage is insufficient in Brazil due to different factors, such as feelings, beliefs, attitudes, socioeconomic aspects, and accessibility [8].

Evidence indicates that, when performed in women under 25 years of age, the test is less efficient, therefore, it is not recommended and has no impact on reducing the incidence and mortality of cervical cancer (INCA, 2016) [4]. The low adherence and the gaps in the population’s knowledge, combined with the opportunistic screening pattern that predominates in Brazil, have generated inadequate demands [8,9].

There is a failure in the effective implementation of the national guidelines for the early detection of cancer, which, according to Santos et al. (2019) [10], is related to the low organizational tradition in the use of the guidelines, low acceptance by professionals, and lack of organization of health services. Therefore, cervical cancer continues to be an important public health problem in Brazil.

According to data from the National Cancer Institute (INCA), this neoplasm is the fourth most common in the female population, with a predominance in the age group of 45 to 50 years. In addition, it represents the fourth leading cause of female mortality [11,12]. A total of 17,010 new cases are estimated for each year of the 2023–2025 triennium, with an estimated risk of 15.38 cases per 100,000 women [2,13].

In this sense, health indicators are important management tools that enable the evaluation of the services offered, measuring the reach of the target population, the supply and access to cervical cancer cytological testing [4].

With the emergence of COVID-19, caused by the novel coronavirus SARS-CoV-2, its high transmissibility and rapid spread of infection, the need for contact restrictions and reallocation of health resources to the front line in combating and treating COVID-19 has arisen. According to the technical notes (NT/INCA, 30 March 2020 and NT/INCA, 9 July 2020) published during the pandemic, new procedures were adopted with restrictions on CCU screening for cervical cancer, which could have postponed diagnoses and the start of necessary care. According to the Brazilian Society of Clinical Oncology (SBOC), evasion of cancer diagnoses and treatments was identified in approximately 75% of the population during the pandemic in 2020 [14].

The lack of cervical cancer screening, often accompanied by the spread of misinformation, has become an even more pressing challenge during the COVID-19 pandemic. Fear of spreading the virus and misinformation about medical procedures may be contributing to women’s reluctance to undergo screening tests. Additionally, online platforms and social media, which are often used as sources of information, can misrepresent myths and spread inaccurate information about the efficacy and safety of tests, which can negatively influence women’s health decisions [15].

The relationship between cervical cancer screening practices, the spread of misinformation, and mental health problems during the pandemic is complex and multifaceted. The widespread uncertainty, social isolation, and financial worries associated with the health crisis can have a significant impact on women’s mental health, leading to symptoms of anxiety and depression. The pandemic and misinformation about reproductive health are exacerbating these issues, creating an environment in which women feel powerless and ignorant, which can lead to poorer mental health in the midst of a global health crisis [13,16].

The likely delay in the diagnosis and treatment of cervical cancer in patients with an abnormal Pap smear during the COVID-19 pandemic will possibly result in the need for secondary prevention measures. These reflexes will have a greater impact in places where people seek the health system for other reasons, as is the case in Brazil. The objective of this study was to analyze the trend of advancement of cervical cancer coverage actions in Brazilian capitals from 2016 to 2021.

## 2. Methods

### 2.1. Type of Study, Period and Location

This is an ecological study with temporal analysis that used secondary data referring to process indicators for cervical cancer control actions in Brazilian capitals between 2016 and 2021 (Discipline of Gynecology, Department of Obstetrics and Gynecology, School of Medicine of the University of São Paulo—FMUSP).

### 2.2. Data Collection Tools and Procedure

The definition of malignant neoplasm of the cervix followed the tenth revision of the International Classification of Diseases (ICD10), code C53, which corresponds to malignant neoplasm of the cervix in the ICD-10 morbidity list. As shown in Figure 1, aspects related to the quality of care in the cervical cancer prevention program were evaluated using the databases of the Cancer Information System—SISCAN (cervical and breast), available on DATASUS (website http://datasus.saude.gov.br/, accessed on 1 August 2022).

The performance of the indicators agreed for the cervical cancer screening program was based on the process and outcome components. Performance indicators determined as indirect measures of quality were used in a monitoring instrument to identify processes or services that need more direct evaluation.

For the present study, the indicators used to monitor and evaluate cervical cancer control actions were (i) Pap smear coverage of the cervix, (ii) the reason for cervical examination, and (iii) the proportion of cervical examinations every three years.

(i) As shown in Figure 1, the provision of coverage and adherence to the national guidelines for technical examinations was analyzed using the indicator of coverage of cervical cytopathological examinations in women in the target population aged 25 to 64 years. This corresponds to the percentage of women in the target population aged between 25 and 64 years, living in a given place and year, who underwent cytopathological examination of the cervix. This indicator contributed to assess the comprehensiveness of cervical cancer prevention actions through screening in the target population.

(ii) As shown in Figure 2, to evaluate the supply of preventive exams for cervical cancer in the female population, we used the indicative ratio of cervical cytopathological tests in women aged 25 to 64 years and in the female population of the same age group. This made it possible to analyze temporal variations in access to this test. The indicator expresses the need to carry out an examination every three years, in accordance with national guidelines.

(iii) As shown in Figure 3, in order to analyze the adequacy of the preventive examination with the recommended periodicity (triennial), the proportion of cytopathological examinations of the uterine cervix with a frequency of three years was evaluated. This indicator evaluates the number of tests performed every three years among the total number of tests performed in the target group.

### 2.3. Data Analysis

The raw values of cervical Pap smears performed before and during the pandemic are presented in a line graph to make temporal understanding clearer. For trend analysis, the Prais-Winsten generalized linear analysis model was used, in which the independent variables (X) were the years in which hospitalizations occurred and hospitalization rates were considered dependent variables (Y). Initially, the logarithmic transformation of the Y values was performed, followed by the application of the Prais-Winsten autoregressive model to estimate the β values and the standardized hospitalization rate in general and by sex. Subsequently, β values corresponding to each of the rates were applied to the following formula to identify the rate of change—APC = [−1 + and β] × 100%.

The confidence level adopted was 95%, and the statistical program used was Data Analysis and Statistical Software for Professionals (Stata) version 16.0.

### 2.4. Ethics Committee

The study was submitted to and approved by the Ethics Committee of the Department of Obstetrics and Gynecology of the Medical School of the University of São Paulo-CEPDOG.

## 3. Results

According to Table 1, in 2016, 410,000 tests were performed and notified in the SISCAN system in all Brazilian capitals, with emphasis on Curitiba, with 65,715 tests performed, and Porto Velho, with 174. In 2019, the capital with the most tests performed was Belo Horizonte (*n* = 79,240), and the one with the fewest tests was Goiânia (*n* = 606). In 2020, there was a reduction in exams compared to the previous year in all capitals, with the exception of Palmas, which went from 7655 exams to 9604.

Table 2 shows that all the capitals studied showed an increase in the annual percentage variation of Pap smear coverage, although Brasília, Manaus, Porto Alegre and Porto Velho did not show a statistically significant increase (APC = 3.01, 2.746, 3.987 and 3.69, respectively).

When analyzing the annual percentage variation in the ratio of cervical oncotic cytology exams performed every 3 years in the target age group, all the capitals studied obtained an increase in the annual percentage variation of oncotic cytology coverage; however, Brasília, Manaus, Porto Alegre and Porto Velho did not show a statistically significant increase (APC = 0.09, 0.082, 0.11 and 0.11, respectively) (Table 3).

Figure 2 and Table 4 documents the number of oncotic cytology tests in 2019, 2020, and 2021, with emphasis on Belo Horizonte, which had the highest number of tests in 2019 (*n* = 79,240) and the third highest in the post-pandemic period, 2021, *(n* = 58,688). In addition, the capital Manaus also stands out, with less than 6730 exams performed in 2019 and 2020 compared to the number of 75,575 exams performed in 2021.

## 4. Discussion

In the period from 2016 to 2021, there was a stable supply of cervical cytopathological tests in the Unified Health System (SUS), with a decrease at the end of the period. The drop in exams in 2020 was a result of the COVID-19 pandemic. In 2021, there was an increase in the number of exams compared to 2020, but still below the levels reached in the years before the pandemic [4].

The indicative ratio of colpocitola in women aged 25 to 64 years and in the female population of the same age group represents proximity to the coverage of the test. Knowledge of this relationship contributes to the clarity between supply and demand and enables the identification of disparities, in addition to subsidizing processes of planning, management and evaluation of public policies in the area of women’s health [16].

Cervical cytological coverage in Brazil is directly linked to socioeconomic and regional inequalities in the country. Social and economic differences are important in determining access to preventive health services, such as Pap smears. Women in disadvantaged socioeconomic situations often face obstacles, such as lack of money, mobility difficulties, and lack of information, which can lead to lower participation in screening programs. These inequalities can be aggravated by cultural and educational issues, directly affecting awareness and acceptance of Pap smears as an effective prevention measure [13,17].

Regional disparities in Brazil play a marked role in cervical cytological coverage. More remote or economically disadvantaged regions often face structural limitations, such as a shortage of health facilities, a lack of trained professionals, and logistical difficulties, which result in less accessibility to screening services. The lack of adequate infrastructure in these regions may contribute to an unequal distribution of Pap smear benefits, increasing gaps in cervical cancer prevention [17].

The stability observed in cytopathological coverage rates is an important indicator of the consistency of cervical cancer coverage activities in Brazilian state capitals [18]. The stationary trend suggests that the awareness-raising strategies and public health programs implemented have been effective in maintaining a constant level of women’s participation in early detection activities. For Lopes and Ribeiro (2019) [19], it is essential to continue monitoring and adapting these strategies to address emerging challenges and ensure that cytopathological coverage remains effective over time.

The increasing trend in cytopathological coverage is a reflection of the success of awareness strategies, public health policies, and collaborative efforts to promote the early detection of cervical cancer [20]. The growing awareness of the importance of screening tests, combined with greater access to health services, has contributed to this positive evolution [21].

The present study demonstrated an increase in screening rates, which contradicted the findings of Vieira et al. (2022) [22], who found in their research that the prevalence of coverage for preventive cervical cancer exams in Brazil increased from 83% in 2011 to 82.2% in 2020, with a trend of increasing coverage in the country in 10 years (2011 to 2020) and an annual variation of 4.6% (*p*-value < 0.001). 

In addition to offering and performing the Pap smear, Santos et al. (2019) [9] considered that, in order to increase Pap smear coverage, it is necessary for the Family Health Strategy (FHS) nurse to actively seek out women, promote bonding and health education, and liaise with community representatives. Estudo argues that adequate nursing consultation is essential for the prevention and early detection of cancer, in addition to representing an important ally of the screening program due to its educational potential [23].

In addition, it is necessary to consider the reasons that influence women not to undergo the test. In this sense, it was observed that the lack of knowledge about the importance of performing preventive exams for colon cancer screening is directly linked to non-adherence to the exam. Lack of time, shame, lack of interest, difficulty in accessing health services, fear of the procedure and the discovery of cancer are factors that hinder the performance of the procedure [24].

For Santos et al. (2012) [25], low proportions may denote a lower capacity of the care network to capture and track and, in some cases, may indicate an inadequate supply of tests, low awareness capacity, and geographic difficulties in accessing services, for example. On the other hand, they can demonstrate good coverage of the program, which results in low adherence of women without previous cytology. Therefore, it is necessary to relate this indicator to the coverage in each capital. In capitals where coverage is low, a progressive increase is expected, due to the recruitment of women in the target age group who will take the exam for the first time [9,13].

The Pap smear every 3 years refers to the number of tests performed every 3 years among the total number of tests performed in the target group. In disagreement with the recommendation to perform the Pap smear every 3 years, studies have indicated a high number of tests performed at intervals of 1 year or less [26]. For Lopes and Ribeiro (2019) [19], the lack of knowledge about the appropriate periodicity constitutes a barrier to its compliance; They also reported that socioeconomic and demographic disparities determine the coverage and adequate frequency of preventive screenings.

The low coverage of Pap smears during the pandemic reflects a number of interconnected challenges, with health infrastructure being one of the determining factors in this scenario. The overload of the health system in response to the pandemic, with the need to redirect resources to fight COVID-19, may have led to a reduction in the capacity to perform preventive tests, such as Pap smears. Limitations in personnel, equipment, and physical space may have contributed to the interruption or delay in the provision of these services, resulting in a significant decrease in cervical cancer screening coverage [14].

Public policies play a crucial role in determining Pap smear coverage during the pandemic. Measures such as lockdowns, mobility restrictions, and resource reallocation may have directly impacted the availability and affordability of these services. Policies that do not adequately prioritize the continuity of preventive care can result in gaps in cervical cancer screening [27]. Patients’ behavior is also influenced by these policies and the uncertainty generated by the pandemic, leading many women to postpone or avoid routine exams for fear of exposure to the virus in hospital settings [3].

During the pandemic, a number of factors influenced health screening, showing the relationship between public health measures, access to health care, patient attitudes, and availability of screening services. Public health actions taken to contain the spread of COVID-19, such as lockdowns, social distancing and redistribution of resources, have directly affected the functioning of tracing services. The reallocation of resources to meet the urgency of the pandemic may have led to the temporary suspension of screening services, making it difficult to continue preventive examinations [28].

Mobility restrictions, fear of exposure to the virus, and a preference for virtual consultations have changed the way doctors and patients interact. These changes have affected patients’ attitudes toward screening tests, resulting in hesitancy or postponement in seeking preventive care. In addition, the lack of screening services due to the redistribution of resources and changes in clinical practices has worsened access [24].

The comparative analysis of the 3-year period of 2019, 2020 and 2021 revealed interesting patterns in the rates of oncotic cytology examinations. While 2019 saw a steady increase in the number of exams, 2020 saw a notable drop. This can be attributed to the impact of the COVID-19 pandemic, which has resulted in mobility restrictions, temporary closures of medical facilities, and concerns about exposure to the virus. The 2021 analysis showed a gradual recovery, indicating a progressive adaptation to the challenges posed by the pandemic [29]

Data analysis indicates that the pandemic has had a substantial impact on cancer cytology coverage. Mobility restrictions, fear of contagion, and the reorganization of health services have affected the demand and availability of tests. The interruption of regular services during lockdowns and the redirection of resources to fight the pandemic may have contributed to the decrease in coverage, especially among vulnerable groups [30].

This comparative analysis highlights the sensitivity of screening rates to external factors, such as public health crises. The fluctuations observed over the three years highlight the importance of adaptive strategies to ensure continuity of cervical cancer prevention in challenging situations [31]. According to the World Health Organization (WHO), it is essential to invest in targeted awareness campaigns, expand access to telemedicine services, and implement innovative approaches to maintain coverage coverage. These strategies are crucial for minimizing the impact of disruptive events on women’s health and maintaining progress in the early detection of cervical cancer [12].

The analysis of Pap smear coverage in Brazilian capitals, considering the COVID-19 pandemic, highlights the complex interaction between public health, social inequalities and adaptation in times of crisis. For Murewanhema (2021) [32] and Poljak et al (2021) [30], the pandemic has brought to light the importance of resilient health systems, comprehensive health education, and adaptive approaches that ensure the ongoing prevention of cervical cancer. The studies proposed by Aarestrup (2022) [33] and Santos Mattos (2023) [34] demonstrate that it is possible to face future health crises more effectively, ensuring women’s health and well-being, regardless of the circumstances.

Ecological studies, despite offering valuable population insights, face challenges such as ecological fallacy, lack of consideration of individual variability, and difficulties in causal inference, limiting the generalization and validity of results.

This study has limitations due to the use of secondary data from Health Information Systems. It is noteworthy that, in SISCAN, “the reliability of the information generated depends on the quality of the data collected and recorded” [12]. “Considering that the SIA maintains several recording instruments, the level of detail of the information available in this system will depend directly on the instrument in which the procedures are recorded” [6].

During crises such as the COVID-19 pandemic, it is necessary to implement adaptive and resilient strategies to improve cervical cytology coverage. Educational campaigns are essential to clarify misconceptions, dispel concerns, and encourage women’s active participation in screening programs [35]. These campaigns can be carried out online through platforms, social media, and other communication channels. In addition, telemedicine facilitates access to medical care, allowing virtual consultations for counseling, clarification of doubts, and even remote exams. Strategic partnerships with community organizations and local health centers are important to address the individual needs of different populations [36].

## 5. Conclusions

This ecological analysis revealed a worrying drop in the number of tests performed in 2020, reflecting the sharp drop in coverage actions in Brazilian capitals during the pandemic caused by the SARS-CoV-2 virus. The pandemic has exacerbated existing inequalities and highlighted the need for adaptive strategies to maintain essential screening services in times of crisis.

## Data Availability

All data critical to the research presented in this article are available for review and validation. Data related to this research can be accessed through the link [https://datasus.saude.gov.br/acesso, 1 August 2022]. The availability of this data aims to promote transparency, replicability, and collaboration in scientific research. I am committed to providing open access to the underlying data from this study to allow for more detailed evaluation and use by other researchers. Any questions or requests related to the data can be directed to my email address: [annielson.costa@usp.br].

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
