# Peer review of "Pap Smear Cancer Coverage in Brazilian Capitals including the Pandemic Period Caused by the SARS-CoV-2 Virus: Ecological Study"

_ijerph, 2024, doi:10.3390/ijerph21030303_

Round 1

Reviewer 1 Report

Comments and Suggestions for Authors

de Souza Costa Costa et al. present a study where they investigated the rates of cervical cancer screening due to the COVID19 pandemic. They highlight that the rates of cervical screening reduced all across Brazil during the pandemic except in certain areas. The findings are unsurprising given the negative impact the COVID19 pandemic has had on preventative as well as curative healthcare. 

Specific comments:

1. The authors may have introduced a typo in the author list and need to correct it.

2. The authors list a number of factors associated with the development of cervical cancer first, and mention HPV infection, which is the most widely established risk factor of cervical cancer later in the abstract.

3. The authors have not detailed the current screening schedule followed in Brazil, whether HPV vaccination is being administered, or how the screening uptake has been since the introduction.

4. In the methods, the authors call this study 'ecological', where they might have meant 'epidemiological'.

5. Did the authors correct for deaths due to COVID19 in their target population?

6. The presented analysis clumps all women 25 - 64 years of age together. More insights could have been derived by dividing the target group into decades.

7. The authors do not discuss specific causes of dropping out of screening, e.g. spread of misinformation, mental health issues developed during pandemic etc.

Comments on the Quality of English Language

 Extensive editing of English language required

Author Response

REVIEWER 01 -

I would like to express my sincere thanks for the detailed review of my manuscript entitled "Coverage of Oncotic Cervical Cytology in Brazilian capitals including the pandemic period caused by the SARS-CoV 2 virus: Ecological Study", submitted to the International Journal of Evironmental Research and Public Health - IJERPH. I appreciate the time and efforts put in by the reviewers and editorial staff.

I am satisfied with the observations and suggestions provided by the reviewers, which have undoubtedly significantly improved the quality of my work. Below, I present my responses to the suggested corrections:

ANSWERS

  1. Thank you. List of authors has been corrected.
  2. Yes, we have listed the factors associated with the development of cervical cancer, mainly emphasizing HPV virus infection.
  3. Thank you, we have included the following suggestions:

The quadrivalent vaccine was added to the national SUS vaccination schedule in March 2014. Currently, the HPV vaccine is available for girls ages 9 to 14 and boys ages 11 to 14. The vaccination schedule consists of two doses, administered 6 months apart. (Brazil, 2018). People aged 9 to 45 years, with HIV/AIDS, who have undergone organ or bone marrow transplants, and cancer patients, who have a weakened immune system (PNI, 2022).

It is recommended that the interval between doses should not exceed 12-15 months to ensure high antibody production and vaccination effectiveness. However, if adolescents or young people are late with vaccination doses, even if the recommended interval of 12-15 months has passed, they should continue the vaccination schedule when they go to the vaccination rooms, without the need to restart (Brasil, 2018).

  1. An ecological study is a type of epidemiological research that looks at associations between variables at the population level, using aggregated data from groups, communities, or populations rather than individual data. In this type of study, the units of analysis are collective, such as countries, states, cities, or demographic groups, and the conclusions are applied to the population as a whole. This approach is based on observing patterns or trends between variables at different locations or times.
  2. We cite the deaths from covid-19 in the entire Brazilian population, with updated data.
  3. The methodological approach chosen for the present study was to group all women between 25 and 64 years of age.
  4. Thanks, we have included the following suggestions:

The lack of cervical cancer screening, often accompanied by the spread of misinformation, has become an even more pressing challenge during the COVID-19 pandemic. Fear of spreading the virus and misinformation about medical procedures may be contributing to women's reluctance to undergo screening tests. Additionally, online platforms and social media, which are often used as sources of information, can misrepresent myths and inaccurate information about the efficacy and safety of tests, which can negatively influence women's health decisions (RIBEIRO, 2022).

The relationship between cervical cancer screening practices, the spread of misinformation, and mental health issues during the pandemic is complex and multifaceted. The widespread uncertainty, social isolation and financial worries associated with the health crisis can have a significant impact on women's mental health, leading to symptoms of anxiety and depression. The pandemic and misinformation about reproductive health are exacerbating these problems, creating an environment in which women feel powerless and ignorant, which can lead to poorer mental health amid a global health crisis (DUARTE, 2022).

-

I am confident that the proposed amendments have substantially improved the clarity and precision of my work. I have attached the revised version of the manuscript, highlighting the modifications made to facilitate revision.

I thank you again for the opportunity to submit my work to the International Journal of Environmental Research and Public Health - IJERPH and for the attention devoted to my research. I look forward to the continuation of the process and, if necessary, I am available to provide additional clarification or make further modifications as directed by the editorial team.

Thank you in advance for the consideration of my response to the corrections and for the work dedicated to promoting academic excellence.

Best regards,

Annielson de Souza Costa

Reviewer 2 Report

Comments and Suggestions for Authors

As a reviewer, I have carefully examined the scientific article titled "Coverage of Oncotic Cervical Cytology in Brazilian capitals including the pandemic period caused by the SARS-CoV-2 virus: An Ecological Study". The study investigates the trends in cervical cytology coverage in Brazilian capitals, considering the impact of the COVID-19 pandemic. Overall, the authors have addressed an important topic and provided relevant findings. However, there are several strengths and weaknesses that should be taken into consideration:

Strengths:

1. Relevance of the Topic: The study addresses an important aspect of public health, particularly the impact of the COVID-19 pandemic on cervical cytology coverage. This topic has significant implications for women's health, especially during times of crisis.

2. Comprehensive Data Analysis: The authors have conducted a thorough analysis of the cervical cytology coverage data for Brazilian capitals from 2016 to 2021. The examination of trends and variations across different cities provides valuable insights into the impact of the pandemic on screening activities.

Weaknesses:

1. Lack of Methodological Details: The article lacks sufficient information regarding the study design, sampling technique, and data collection methods. This hinders the replicability and transparency of the study.

2. Limited Discussion of Results: The authors primarily present the descriptive findings but fail to provide a comprehensive analysis and interpretation. The discussion section should further explore the reasons behind the observed changes in coverage rates during the pandemic and highlight potential implications.

3. Insufficient Focus on Inequalities: Although the study mentions inequalities in coverage, it fails to delve into a detailed analysis of socio-economic factors, regional disparities, or barriers preventing women from accessing cervical cytology. A more nuanced examination of these factors would strengthen the study's impact.

Here are some suggestions for improving the manuscript by adding or clarifying certain information:

1. Methodological Transparency: The authors should provide detailed information on the study design, sampling approach, and data collection methods. This would enable better understanding and evaluation of the study's validity and reliability. Clarify Sampling and Data Collection: Provide additional details on the sampling strategy employed to select the Brazilian capitals for inclusion in the study. Additionally, specify the sources from which the cervical cytology coverage data were obtained, including any limitations or potential biases associated with the data sources.

2. In-depth Analysis of Findings: The authors should expand the discussion section to provide a comprehensive analysis of the observed trends, including potential reasons behind the variations in coverage rates during the pandemic. This could involve exploring the impact of healthcare infrastructure, public health policies, and patient behavior.

3. Addressing Socioeconomic and Regional Inequalities: A deeper investigation of socioeconomic determinants and regional disparities would enhance the study's relevance and contribute to a more comprehensive understanding of the barriers hindering cervical cytology access. This analysis could include factors such as income, education, healthcare facilities, and geographical location.

4. Enhance Description of Variables: Clearly define the variables used in the study, such as "cervical cytology coverage" and "reasons for cervical cytology." Explain how these variables were measured or calculated, including any specific criteria or thresholds applied.

5. Discussion of Results: Expand on the discussion of the findings to include potential explanations for the observed trends in cervical cytology coverage during the pandemic. Consider addressing factors such as public health measures, changes in healthcare access, patient attitudes or concerns, and the availability of screening services. This would enrich the interpretation of the results and provide more context for the observed changes.

6. Inclusion of Limitations: Explicitly acknowledge the limitations of the study, including any potential sources of bias or confounding. Discuss the impact of these limitations on the validity and generalizability of the findings. This will enhance the transparency and credibility of the study.

7. Implications and Recommendations: Provide a section discussing the implications of the findings and suggesting potential strategies for improving cervical cytology coverage, especially during times of crisis. Discuss potential interventions or policies that could address the observed disparities and ensure access to screening services for all population groups.

By incorporating these suggestions, the manuscript would provide a more comprehensive and informative analysis of the cervical cytology coverage in Brazilian capitals, including the impact of the pandemic.

Comments on the Quality of English Language
Average

Author Response

REVIEWER 02-

I would like to express my sincere thanks for the detailed review of my manuscript entitled "Coverage of Oncotic Cervical Cytology in Brazilian capitals including the pandemic period caused by the SARS-CoV 2 virus: Ecological Study", submitted to the International Journal of Evironmental Research and Public Health - IJERPH. I appreciate the time and efforts put in by the reviewers and editorial staff.

I am satisfied with the observations and suggestions provided by the reviewers, which have undoubtedly significantly improved the quality of my work. Below, I present my responses to the suggested corrections:

ANSWERS

  1. Study design/sampling approach: The present research is an ecological study, with temporal analysis that used secondary data related to process indicators for cervical cancer control actions in Brazilian state capitals.

Data collection sampling: The study was carried out with women living in Brazilian capitals (Aracaju, Belém, Belo Horizonte, Boa Vista, Brasília, Campo Grande, Cuiabá, Curitiba, Florianópolis, Fortaleza, Goiânia, João Pessoa, Macapá, Maceió, Manaus, Natal, Palmas, Porto Alegre, Porto Velho, Recife, Rio Branco, Rio de Janeiro, Salvador, São Luís, São Paulo, Teresina and Vitória) aged 25 to 64 years attending health establishments with a public legal nature or private sector (provides services to the Unified Health System on a complementary basis).

The source of cervical cytology coverage data was obtained from databases of the Cancer Information System – SISCAN (cervix and breast), which is available at DATASUS (website http://datasus.saude.gov.br/).

Limitations: Ecological studies, while providing valuable population-level insights, have important limitations. The ecological fallacy, which involves misinterpreting associations in groups as valid for individuals, poses a significant challenge.

  1. Thank you, we have included the following suggestions:

The low coverage of Pap smears during the pandemic reflects a series of interconnected challenges, with health infrastructure being one of the determining factors in this scenario. The overload of the health system in response to the pandemic, with the need to redirect resources to cope with COVID-19, may have led to a reduction in the capacity to perform preventive tests, such as Pap smears. Limitations in personnel, equipment, and physical space may have contributed to the interruption or delay in the provision of these services, resulting in a significant decrease in cervical cancer screening coverage (RIBEIRO, 2022).

Public policies play a crucial role in determining Pap smear coverage during the pandemic. Measures such as lockdowns, mobility restrictions, and resource reallocation may have directly impacted the availability and affordability of these services. Policies that do not adequately prioritize the continuity of preventive care can result in gaps in cervical cancer screening (FONSECA, 2021). Patients' behavior is also influenced by these policies and the uncertainty generated by the pandemic, leading many women to postpone or avoid routine exams for fear of exposure to the virus in hospital environments (FERREIRA, 2022).

  1. Thank you, we have included the following suggestions:

Cervical cytological coverage in Brazil is directly linked to socioeconomic and regional inequalities in the country. Social and economic differences are important in determining access to preventive health services, such as Pap smears. Women in disadvantaged socioeconomic situations often face obstacles, such as lack of money, mobility difficulties, and lack of information, which can lead to lower participation in screening programs. These inequalities can be worsened by cultural and educational issues, directly affecting the awareness and acceptance of Pap smear as an effective prevention measure (INCA, 2022).

Regional disparities in Brazil play a marked role in cervical cytological coverage. More remote or economically disadvantaged regions often face structural limitations, such as a shortage of health facilities, a lack of trained professionals, and logistical difficulties, which result in less accessibility to screening services. The lack of adequate infrastructure in these regions may contribute to an unequal distribution of the benefits of Pap smear, increasing gaps in cervical cancer prevention. (MANICA, 2016).

  1. Coverage of cervical Pap smears was calculated as the percentage of the number of women aged 25 to 64 years with cervical Pap smears performed in the last three years, living in a given place and year, divided by the percentage of the number of women aged 25 to 64 years, living in the respective place and year.

The ratio of cervical cytopathological examinations was calculated as  the percentage of the number of cervical cytopathological examinations in women aged 25 to 64 years, living in a given location, divided by the percentage of the number of women aged 25 to 64 years, living in the respective location and year/3.

  1. Thank you, we have included the following suggestions:

During the pandemic, a number of factors influenced health screening, showing the relationship between public health measures, access to health care, patient attitudes, and availability of screening services. Public health actions taken to contain the spread of COVID-19, such as lockdowns, social distancing, and redistribution of resources, have directly affected the functioning of tracing services. The reallocation of resources to meet the urgency of the pandemic may have caused the temporary suspension of screening services, making it difficult to continue preventive exams (AQUINO, 2020).

Mobility restrictions, fear of exposure to the virus, and a preference for virtual consultations have changed the way doctors and patients interact. These changes have affected patients' attitudes toward screening tests, resulting in hesitancy or postponement in seeking preventive care. In addition, the lack of screening services, due to the redistribution of resources and changes in clinical practices, has worsened access (SILVA, 2020).

  1.  Ecological studies, despite offering valuable population insights, face challenges such as ecological fallacy, lack of consideration of individual variability, and difficulties in causal inference, limiting the generalization and validity of results.

  1. Thank you, we have included the following suggestions:

During crises, such as the COVID-19 pandemic, it is necessary to implement adaptive and resilient strategies to improve cervical cytology coverage. Educational campaigns are essential to clarify misconceptions, dispel concerns and encourage women's active participation in screening programmes (TRECO, 2021).  These campaigns can be carried out online through platforms, social media, and other communication channels. In addition, telemedicine facilitates access to medical care, allowing virtual consultations for counseling, clarification of doubts, and even remote exams. Strategic partnerships with community organizations and local health centers are important to individually meet the needs of different populations (CAVALHEIRI, 2018).

----

I am confident that the proposed amendments have substantially improved the clarity and precision of my work. I have attached the revised version of the manuscript, highlighting the modifications made to facilitate revision.

I thank you again for the opportunity to submit my work to the International Journal of Environmental Research and Public Health - IJERPH and for the attention devoted to my research. I look forward to the continuation of the process and, if necessary, I am available to provide additional clarification or make further modifications as directed by the editorial team.

Thank you in advance for the consideration of my response to the corrections and for the work dedicated to promoting academic excellence.

Best regards,

Annielson de Souza Costa

Round 2

Reviewer 1 Report

Comments and Suggestions for Authors

I have no further comments

Comments on the Quality of English Language

None